# Characterization of dengue patients in Vietnam: Clinical, virological, and IL-10 profiles during 2021– 2022 outbreaks

Do Duc Anh[1,2], Lara Vugrek[1], Nguyen Trong The[2,3], Nourhane Hafza[1], Truong Nhat My[2], Le Thi Kieu Linh[1,2], Do Huy Loc[1,2], Jonas Schmidt-Chanasit[4,5], Nguyen Linh Toan[2,6], Peter G. Kremsner[1,7], Le Huu Song[2,3], Thirumalaisamy P. Velavan [1,2,8]*

1 Institute of Tropical Medicine, University of Tübingen, Tübingen, Germany, 2 Vietnamese-German Center for Medical Research (VG-CARE), Hanoi, Vietnam, 3 108 Military Central Hospital, Hanoi, Vietnam, 4 Bernhard Nocht Institute for Tropical Medicine, Hamburg, Germany, 5 Faculty of Mathematics, Informatics and Natural Sciences, University of Hamburg, Hamburg, Germany, 6 Vietnam Military Medical University, Hanoi, Vietnam, 7 Centre de Recherches Médicales de Lambaréné (CERMEL), Gabon, 8 Faculty of Medicine, Duy Tan University, Da Nang, Vietnam

* t.velavan@uni-tuebingen.de

## Abstract

### Background

The pathogenesis of dengue is attributed to a complex interaction between the dengue virus (DENV) and the host immune system. The aim of this study is to investigate the clinical, virological, and Interleukin-10 (IL-10) profiles of dengue patients in Vietnam from two consecutive outbreaks in 2021 and 2022.

### Methods

A total of n=306 dengue patients were examined, who were clinically stratified according to dengue without warning signs (DF; n=178), dengue with warning signs (DWS; n=115) and severe dengue (SD; n=13). Patients were screened for dengue, Zika and chikungunya viruses. DENV were subjected to serotype specific real-time RT-PCR. Interleukin-10 (IL-10) levels were measured by ELISA, and *IL-10* promoter variants (-1082G/A; -819C/T; -592C/A) were genotyped by direct Sanger sequencing to determine a possible association with susceptibility to dengue and disease severity.

### Results

No chikungunya or Zika viruses were detected. Patients were infected by one of the three different DENV serotypes (DENV-1, -2, -4). Plasma IL-10 levels were significantly elevated in patients (DF vs. DWS, p=0.004; DF vs. SD, p=0.001; DWS vs. SD, p=0.015). While the *IL-10* allele -819C contributed to an increased risk of dengue (OR = 1.5, 95% CI = 1.1-2.0, p=0.04), genotype -1082GA showed a protective role against the disease (OR = 0.45, 95% CI = 0.27-0.72, p=0.009), and allele -1082G showed a protective role against DWS (OR = 0.44, 95% CI = 0.22-0.81, p=0.049). Also, the *IL-10* GTA (-1082G/-819T/-592A) haplotype was observed to confer protection (OR = 0.31, 95% CI = 0.14-0.67, p< 0.003).

**Data availability statement:** The authors confirm that the data supporting the findings of this study are available within the article.

**Funding:** TPV acknowledges the funding from the PAN-ASEAN Coalition for Epidemic and Outbreak Preparedness (PACE-UP; German Academic Exchange Service (DAAD) Project ID: 57592343). The funder has no role in the study design, data collection and analysis, decision to publish, or preparation of the manuscript.

**Competing interests:** The authors have declared that no competing interests exist.

## Conclusion

While DENV-1 and DENV-2 were the predominant serotypes in circulation, plasma IL-10 levels and *IL-10* promoter variants were also significantly associated with dengue and its severity.

## Author summary

Dengue places a significant burden especially in low- and middle-income countries, including Vietnam. The pathogenesis of dengue is attributed to a complex interaction between the dengue virus and the host immune system. We explicitly investigated on the occurrence of arboviruses (CHIKV, DENV and ZIKA) during two consecutive out-breaks in 2021 and 2022 from Northern Vietnam and compared the results with those of previous outbreaks. From the study cohort of cases of dengue without warning signs, dengue with warning signs and severe dengue, we investigated whether levels of the human cytokine IL-10 and *IL-10* promoter variants were associated with susceptibility and with their clinical course. The results show that DENV-1, DENV-2 and DENV-4 were the circulating serotypes in 2021-2022. Serotypes DENV-1 and DENV-2 were more frequently observed in dengue cases with warning signs and in severe dengue, while no cases of chikungunya or Zika viruses were detected in the studied period. Plasma IL-10 levels and *IL-10* promoter variants were significantly modulated in patients with varying degrees of severity.

## Introduction

Dengue, caused by dengue virus (DENV), is a mosquito-borne viral disease that is a major public health problem in Southeast Asia. While the Americas and Asia carry >70% of the global dengue burden, the number of dengue cases has increased more than twofold in the last decade, from 2010 to 2019 [1]. Dengue is caused by one of the four genetically distinct sero-types (DENV-1,2,3 and -4) and the pathogenesis is influenced by a complex interplay between DENV serotypes, host factors and differential host immune responses.

While infections with DENV-2 have a higher tendency to develop severe dengue than other serotypes [2,3], host factors such as innate and adaptive immunity, antibody-dependent enhancement (ADE), cross-reactive memory T cells are also important determinants of den-gue severity [4,5]. DENV-2 and DENV-3 were observed to elicit a stronger cytokine response than other serotypes [6], these differences in immunogenicity and pathogenesis are associated with distinct DENV serotypes.

Endothelial dysfunction leading to vascular leakage is the hallmark of severe dengue [7]. It is known that altered cytokine levels are associated with increased endothelial cell damage and plasma leakage [8]. Previous studies have reported that DENV infection can induce the production of interleukin-10 (IL-10) by monocytes, which dampens anti-DENV immune responses and virus control [9,10]. While elevated IL-10 levels have been shown to be asso-ciated with severe dengue [11] and to be a predictive marker for secondary DENV infection [12], the *IL-10* promoter variants rs1800872 (-592C/A) and rs1800871 (-819C/T), located on chromosome 1q31-32, have been shown to dysregulate IL-10 serum levels [13].

Population-specific genetic variations in *IL-10* are known to be associated with susceptibil-ity and varying clinical courses of dengue. Therefore, the study of IL-10 levels and its genetic

variants in the context of dengue is important to decipher the complex interplay between the host immune response and the DENV, and similarly provide insights about *IL-10* variants in the Vietnamese population. In this context, firstly, dengue patients with varying degrees of severity from two consecutive seasonal outbreaks (2021 and 2022) were screened for dengue, Zika and chikungunya viruses in this study cohort and DENV serotypes were studied. Secondly, IL-10 plasma levels were quantified, and *IL-10* variants were genotyped to determine a possible association with susceptibility to dengue and disease severity.

## Materials and methods

### Ethics statement

Written and signed informed consent was obtained from all study participants prior to enrolment. The study was approved by the Institutional Review Board of the 108 Military Hospital and by the University of Tübingen for the project 'Host and viral factors influencing dengue severity and susceptibility' (Ethics Approval No. 274/2022B02). The study has adopted and implemented the Nagoya Protocol and received approval for the utilization of genetic resources in Germany from the Vietnamese Ministry of Natural Resources and Environment (Reference: No.2995/QĐ-BTNMT).

### Study premise and study population

The study employed a convenience sampling method, including patients who were admitted to the central hospital in Hanoi, Vietnam, during two consecutive seasonal dengue outbreaks between October 2021 and December 2022. A total of 306 civilian patients with symptoms of haemorrhagic fever admitted to the 108 Military Central Hospital in Hanoi, Vietnam, participated in the study. Patients with bacterial or other viral infections, chronic diseases or haematological disorders were excluded. The infection was diagnosed based on the diagnostic criteria for dengue according to the World Health Organization (https://apps.who.int/iris/handle/10665/44188) approved by the Vietnamese Ministry of Health [14] and the positivity of NS1 antigen or/and anti-DENV immunoglobulin M and G (anti-DENV IgM and IgG). Serological tests for dengue infection were carried out on admission. Blood samples from all dengue patients were collected at admission and plasma samples were separated and stored at -70°C. In addition, 200 μL of whole blood was collected in QIAcard FTA Indicating Mini (Qiagen GmbH, Hilden, Germany) and stored at room temperature. Similarly, blood samples from 300 healthy blood donors who tested negative for HBsAg, anti-HCV and anti-HIV were collected from the transfusion department.

### Patients clinical parameters

Patients were categorized into three groups based on the 2009 WHO guidelines: Dengue without warning signs (DF), dengue with warning signs (DWS) and severe dengue (SD). Measurements of aspartate aminotransferase (AST), alanine aminotransferase (ALT), white blood cell (WBC) count, red blood cell (RBC) count, haematocrit (HCT), platelet count (PLT) and dengue diagnosis were conducted in real time at 108 Military Central Hospital in Hanoi.

### Screening for dengue, Zika and chikungunya viruses

Total RNA was extracted from 140μL patient plasma using QIAmp Viral RNA Mini Kit (Qiagen GmbH, Hilden, Germany) following the manufacturer's protocol. To exclude other arboviruses and confirm dengue infection, all samples (n=306) were screened for dengue/Zika/chikungunya by multiplex real-time PCR using the Fast Track Diagnostics Kit (Siemens

Healthcare GmbH, Erlangen, Germany) on a LightCycler480-II (Roche, Mannheim, Germany) according to the manufacturer's instructions. All samples were tested in duplicate to ensure accuracy.

### Dengue virus serotyping

All dengue positive patient DENV RNA samples (n=299/306) were serotyped using the RealStar Dengue Type real-time PCR kit 1.0 (Altona Diagnostics GmbH, Hamburg, Germany) following manufacturer's instructions, on a LightCycler480-II (Roche, Mannheim, Germany). All samples were tested in duplicate to ensure accuracy.

### Quantification of interleukin-10

IL-10 levels were quantified from all DENV RNA-positive patient samples (n=299/306) using the Bio-Plex IL-10 Pro Human Cytokine Screening Panel (Bio-Rad Laboratories GmbH, Feldkirch, Germany) according to the manufacturers protocol in a Bio-Plex 200 system and the values were then quantified using Bio-Plex Manager 6.0 software (Bio-Rad Laboratories, Hercules, CA, USA).

### Genotyping of interleukin-10 variants

Genomic DNA was extracted from dried blood spots (n=299) from patients and blood pellets (n=300) from healthy controls using the commercially available QIAamp DNA mini kit (Qiagen GmbH, Hilden, Germany) following the manufacturer's instructions. The quality and quantity of extracted DNA were checked using the NanoDrop (Thermo Fisher Scientific, Waltham, MA, USA). The *IL-10* promoter region containing the polymorphisms rs1800896 (-1082A/G), rs1800871 (-819C/T), and rs1800872 (-592C/A) were amplified by PCR with specific primer pairs IL-10F 5′-GAA GAA GTC CTG ATG TCA CTGC-3′ (forward) and IL-10R 5′-TAG GTC TCT GGC CTT AGT TTC-3′ (reverse) [15].

In brief: PCR reactions were performed in 15μL reaction volume with 5 ng of genomic DNA, 1x HotStarTaq Master mix (Qiagen GmbH, Hilden, Germany), and 0.5 μM of each primer. The thermal cycling parameters were an initial denaturation at 95 °C for 15 minutes, followed by 35 cycles of denaturation (30 seconds at 94 °C), annealing (60 seconds at 62 °C), extension (60 seconds at 72 °C), followed by a final extension at 72 °C for 10 minutes. PCR amplicons were stained with SYBR green and run on a 1.2% gel electrophoresis gel. A ~760 bp product was visualized with a UV transilluminator. PCR products were purified using Exo-SAP-IT PCR (Applied Biosystems, Beverly, MA, USA) and purified amplicons were used as a sequencing template using the BigDye Terminator v.1.1 Cycle Sequencing Kit on an ABI 3130XL DNA sequencer (Applied Biosystems, Beverly, MA, USA). All *IL-10* sequences (~760 bp) were aligned to an *IL-10* reference gene (NG_012088.1) using Bio-edit 7.2 software (https://bioedit.software.informer.com/7.2/) and genotypes were labelled as either homozygous or heterozygous, which was visually confirmed using the respective electropherograms.

### Statistical analysis

Data was analysed and visualized using the R software version 4.3.2 (http://www.r-project.org). A p-value < 0.05 was considered statistically significant. Clinical and demographic data were presented either as median or mean values with range for quantitative variables and absolute numbers with percent for categorical variables. The normality of distribution in the quantitative variables was tested using the Shapiro–Wilk test. Categorical data were compared using Chi-square or Fisher's exact tests, while continuous variables were compared using t-test or Kruskal-Wallis test as appropriate. Dunn's test was applied as post-hoc pairwise tests with

Bonferroni adjustment. The Benjamini-Hochberg Procedure was applied for multiple testing due to the increased risk of type I error with false discovery rate equal to 0.05.

Allele, genotype, or haplotype frequencies were determined by simple gene counting and deviations from Hardy-Weinberg equilibrium were tested. The association between IL-10 genetic variants and dengue was assessed using logistic regression, adjusting for age and sex, under three inheritance models: dominant (comparing homozygotes for the major allele to heterozygotes and homozygotes for the minor allele), recessive (comparing homozygotes for the major allele and heterozygotes to homozygotes for the minor allele), and over-dominant (comparing heterozygotes to homozygotes for both the major and minor alleles). The *IL-10* haplotypes were estimated using the R package "SNPassoc" [16] version 2.1.0 and "haplo. stats" [17] version 1.9.3. The linkage disequilibrium (LD) analysis was performed with the program Haploview v.4.1 (https://www.broadinstitute.org/haploview/haploview).

## Results

### Baseline characteristics of study subjects

The patients and healthy controls were from the Hanoi metropolitan area and were of Kinh ethnicity. Patients were stratified according to the severity of dengue: dengue without warning signs (DF) (n=178), dengue with warning signs (DWS) (n=115) and severe dengue (SD) (n=13). The demographic and clinical characteristics of the patients are summarized in Table 1. No significant differences in age and sex were observed (Table 1). Significant differences were observed in the days of fever before admission, blood parameters, liver enzymes and bleeding manifestations (Table 1). While NS1 positivity was not significant between the groups, IgM and IgG showed significant differences in the distribution between the analyzed groups (Table 1).

### Dengue, Zika and chikungunya detection and dengue virus serotyping

Neither Zika nor chikungunya viral RNA was detected in any of the 306 tested cases. The multiplex PCR assay confirmed that 299/306 cases were positive for DENV RNA. Seven cases (7/306) were not recognized as DENV RNA-positive by the real-time PCR. Of the 299 DENV-positive cases, the DENV serotypes were determined in 280 samples (Fig 1a and Table 1). Three DENV serotypes (DENV-1,-2 and -4) were detected except for the DENV-3 serotype. The prevalence and dominance trend of DENV serotypes remained consistent during the two outbreaks (2021 and 2022), with DENV-2 being the most frequently detected, followed by DENV-1 and DENV-4. Co-infections with DENV-1 and DENV-2 (n=56, 20%) were predominant, followed by DENV-2 and DENV-4 co-infections (n=12, 4%). All three serotypes contributed to the severity of the infection. DENV-1, followed by DENV-2 and DENV-4 contributed to the severe cases. (Table 1).

### Plasma IL-10 levels

Significant differences in IL-10 levels were observed: DF (median = 9.9 pg/mL, range [1.1 – 3060]), DWS (median = 18.0 pg/mL, range [1.4 – 183]) and SD (median = 46.3 pg/mL, range [8.3 – 128]). The distribution of IL-10 levels differed significantly between the study groups (DF vs. DWS, p = 0.004; DF vs. SD, p = 0.001; DWS vs. SD, p = 0.015) (Fig 2).

### *IL-10* variants with DENV infection, severity, and serotypes

All *IL-10* variants analyzed in patients and controls were in Hardy-Weinberg equilibrium. The association of the investigated variants was analyzed using different genetic models, including

**Table 1. Patient characteristics on admission during seasonal dengue outbreaks.**

| Characteristics | Dengue without warning signs (DF) (n=178) | Dengue with warning signs (DWS) (n=115) | Severe dengue (SD) (n=13) | p-value |
|---|---|---|---|---|
| Median age (Range) | 45 [12-86] | 47 [15-82] | 50.5 [19-80] | 0.298** |
| Sex (Male/Female) | 92/80 | 59/56 | 5/7 | 0.71* |
| Days of Fever Mean (±SD) | 3.65 (1.56) | 5.24 (1.33) | 5.25 (0.866) | < 0.001** |
| Days of Fever Median [Range] | 4.00 [1.00, 8.00] | 5.00 [1.00, 8.00] | 5.00 [4.00, 7.00] | < 0.001** |
| Leucocytes/µL Median [Range] | 4.01 [0.930, 16.9] | 3.71 [1.33, 11.6] | 5.00 [1.45, 10.5] | 0.799** |
| Lymphocyte (%) Median [Range] | 22.8 [2.50, 72.8] | 28.3 [6.10, 56.3] | 21.0 [6.80, 53.0] | 0.015** |
| Platelets ×$10^3$/µL Median [Range] | 115 [9.00, 384] | 20.0 [4.00, 228] | 29.0 [4.00, 125] | < 0.001** |
| AST Median U/L [Range] | 54.0 [15.1, 1210] | 115 [16.0, 1040] | 257 [31.0, 11100] | < 0.001** |
| ALT Median U/L [Range] | 38.7 [8.00, 855] | 66.9 [8.20, 636] | 113 [25.6, 2190] | < 0.001** |
| Bleeding manifestation n (%) | 34 (20%) | 97 (84%) | 9 (75%) | < 0.001* |
| Serotypes | | | | |
| DENV-1 n (%) | 37 (22%) | 18 (16%) | 5 (42%) | NA |
| DENV-2 n (%) | 70 (41%) | 71 (62%) | 4 (33%) | NA |
| DENV-4 n (%) | 3 (2%) | 3 (3%) | 1 (8%) | NA |
| DENV-1/2 n (%) | 45 (26%) | 11 (10%) | 0 (0%) | NA |
| DENV-2/4 n (%) | 7 (4%) | 5 (4%) | 0 (0%) | NA |
| Unidentified | 10 (6%) | 7 (6%) | 2 (17%) | NA |
| Serological tests | | | | |
| NS1 – positivity (%) | 127 (74%) | 82 (71%) | 9 (75%) | 0.881* |
| IgM – positivity (%) | 62 (36%) | 74 (64%) | 5 (42%) | < 0.001* |
| IgG – positivity (%) | 56 (33%) | 70 (61%) | 7 (58%) | < 0.001* |

*P-values were calculated by Chi-square test;

**P-values were calculated by Kruskal-Wallis test. Variables were summarized in Percentage, Mean (standard deviation) or Median [range]. AST: Aspartate aminotransferase; ALT: Alanine Aminotransferase; DENV: Dengue virus; NS1: non-structural protein; IgG: immunoglobulin G; IgM: immunoglobulin M.

allelic, dominant, recessive, and over dominant. The variant alleles -1082A, -819T, and -592A were predominant in the Vietnamese population (Table 2). Linkage disequilibrium (LD) analysis revealed a strong LD (D' > 0.90) between -819C/T and -592C/A. The frequency of allele -819C was significantly higher in dengue patients than in healthy controls (OR = 1.5, 95%CI = 1.1-2.0, p-value = 0.04) (Table 2). The frequency of genotype -1082GA was higher in healthy control than in patients (OR = 0.45, 95%CI = 0.27-0.72, p-value = 0.009), suggesting a protective effect (Table 2). The dominant model demonstrated a significant association between genotype -1082 GG+GA and a reduced risk of dengue (OR = 0.5, 95% CI = 0.31-0.8, p-value = 0.024). Conversely, the over-dominant model indicated a positive association between genotype -1082 GG+AA and dengue (OR = 2.28, 95% CI = 1.41-3.71, p-value = 0.009).

### Association of *IL-10* variants with varying severity and DENV serotypes

Significant associations were only observed among DWS patients and controls (Table 3): the frequency of the -1082G allele was found higher in controls compared to DWS patients (OR = 0.44, 95% CI = 0.22-0.81, p-value = 0.049), suggesting that the IL-10 SNP rs1800896 -1082A/G is associated with protection from DWS. Additionally, the recessive model revealed that the genotype -1082GG was associated with protection from developing DWS (OR = 0.4, 95% CI = 0.19-0.77, p-value = 0.049), while the genotype -1082GA was associated with increased risk of DWS under the over-dominant model. No significant difference in IL-10 plasma levels between the *IL-10* variants was observed in our study.

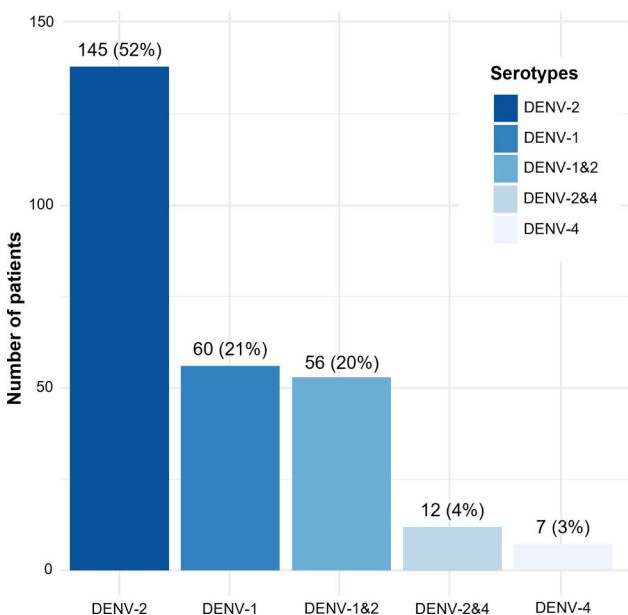

**Fig 1. Distribution of DENV serotypes among patients.** Data are presented as number of observations and percentages.

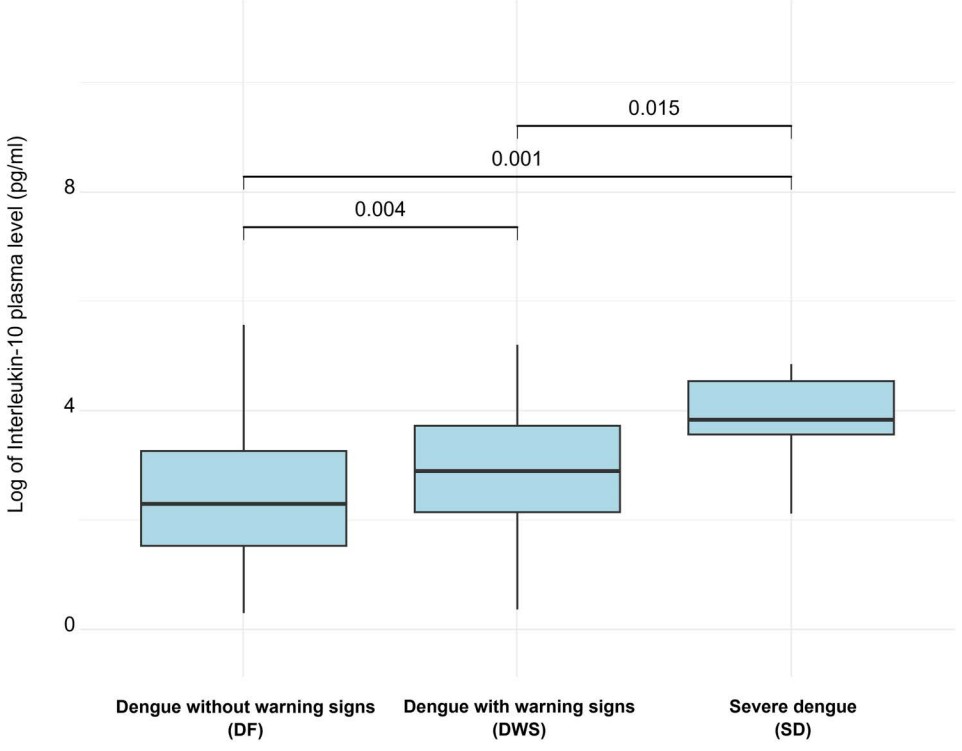

**Fig 2. IL-10 plasma levels in dengue patients with different clinical severities.**

**Table 2. Genotype and allele distribution of *IL-10* polymorphisms in dengue patients and healthy controls. Bold values indicate statistical significance. Odds ratio and p-value were calculated by using binary logistic regression models adjusted for age and sex. All p-values were adjusted by the Benjamini-Hochberg procedure.**

| SNP | Variable | Genotypes | DENV n = 287 | Control n = 282 | OR (95% CI) | p-value |
|---|---|---|---|---|---|---|
| rs1800872 -592C/A | Genotypes | AA | 159 (55.4) | 170 (60.3) | Ref | |
| | | CA | 102 (35.5) | 97 (34.4) | 1.19 (0.81-1.76) | 0.398 |
| | | CC | 26 (9.1) | 15 (5.3) | 1.96 (0.95-4.17) | 0.130 |
| | Allele | A | 420 (73.2) | 437 (77.5) | Ref | |
| | | C | 154 (26.8) | 127 (22.5) | 1.34 (1.04-1.8) | 0.096 |
| | Dominant | AA | 159 (55.4) | 170 (60.3) | Ref | |
| | | CC + CA | 128 (44.6) | 112 (39.7) | 1.32 (0.91-1.91) | 0.195 |
| | Recessive | CA + AA | 261 (90.9) | 267 (94.7) | Ref | |
| | | CC | 26 (9.1) | 15 (5.3) | 1.83 (0.9-3.83) | 0.164 |
| | Over-dominant | CC + AA | 185 (64.5) | 185 (65.6) | Ref | |
| | | CA | 102 (35.5) | 97 (34.4) | 0.88 (0.6-1.29) | 0.527 |
| rs1800871 -819C/T | Genotypes | TT | 160 (55.7) | 178 (63.1) | Ref | |
| | | CT | 101 (35.2) | 91 (32.3) | 1.31 (0.89-1.94) | 0.218 |
| | | CC | 26 (9.1) | 13 (4.6) | 2.28 (1.08-5) | 0.085 |
| | Allele | T | 421 (73.3) | 447 (79.3) | Ref | |
| | | C | 153 (26.7) | 117 (20.7) | 1.5 (1.11-2.04) | **0.04** |
| | Dominant | TT | 160 (55.7) | 178 (63.1) | Ref | |
| | | CC + CT | 127 (44.3) | 104 (36.9) | 1.5 (1.03-2.18) | 0.085 |
| | Recessive | CT + TT | 261 (90.9) | 269 (95.4) | Ref | |
| | | CC | 26 (9.1) | 13 (4.6) | 2.21 (1.06-4.8) | 0.085 |
| | Over-dominant | CC + TT | 186 (64.8) | 191 (67.7) | Ref | |
| | | CT | 101 (35.2) | 91 (32.3) | 0.81 (0.55-1.18) | 0.303 |
| rs1800896 -1082G/A* | Genotypes | AA | 238 (84.4) | 169 (72.2) | Ref | |
| | | GA | 39 (13.8) | 64 (27.4) | 0.45 (0.27-0.72) | **0.009** |
| | | GG | 5 (1.8) | 1 (0.4) | 4.77 (0.63-98.94) | 0.218 |
| | Allele | A | 515 (91.3) | 402 (85.9) | Ref | |
| | | G | 49 (8.7) | 66 (14.1) | 0.61 (0.4-0.94) | 0.085 |
| | Dominant | AA | 238 (84.4) | 169 (72.2) | Ref | |
| | | GG + GA | 44 (15.6) | 65 (27.8) | 0.5 (0.31-0.8) | **0.024** |
| | Recessive | GA + AA | 277 (98.2) | 233 (99.6) | Ref | |
| | | GG | 5 (1.8) | 1 (0.4) | 6.24 (0.84-128.23) | 0.173 |
| | Over-dominant | GA | 39 (13.8) | 64 (27.4) | Ref | |
| | | GG + AA | 243 (86.2) | 170 (72.6) | 2.28 (1.41-3.71) | **0.009** |

*Analysis was performed with 282 patients and 234 control

## Haplotype association

We reconstructed the *IL-10* haplotypes based on three studied SNPs. The frequencies of these observed haplotypes were significantly different between compared groups (Table 5). Data showed that ATA (-1082G/-819T/-592A) is the most frequently observed haplotype in both patient and controls groups (70% of overall haplotypes). The GTA haplotype (-1082A/-819T/-592A) is associated with protection from dengue (OR = 0.31, 95% CI = 0.14 - 0.67, p = 0.003), while the ACC haplotype was found more frequently in dengue patients, and is associated with a higher risk of dengue with borderline significance (OR = 1.36, 95% CI = 0.99 – 1.89, p

**Table 3. Genotype and allele distribution of *IL-10* polymorphisms in dengue patients with different severity and healthy controls. Data are shown in number and percentage. Bold values indicate statistical significance. Odds ratio and p-value were calculated by using binary logistic regression models adjusted for age and sex. All p-values were adjusted by the Benjamini-Hochberg procedure.**

| SNP | Variable | Geno-types | DF n = 162 | Control n = 282 | OR (95% CI) | p-value | DWS n = 113 | OR 2 (95% CI) | p-value | SD n = 12 | OR 3 (95% CI) | p-value |
|---|---|---|---|---|---|---|---|---|---|---|---|---|
| **rs1800872 -592C/A** | Genotypes | AA | 91 (56.2) | 170 (60.3) | Ref | | 61 (54.0) | Ref | | 7 (58.3) | | |
| | | CA | 57 (35.2) | 97 (34.4) | 1.19 (0.76-1.87) | 0.464 | 40 (35.4) | 1.07 (0.62-1.83) | 0.842 | 5 (41.7) | 0.94 (0.19-3.81) | 0.941 |
| | | CC | 14 (8.6) | 15 (5.3) | 1.88 (0.81-4.34) | 0.227 | 12 (10.6) | 2.06 (0.77-5.3) | 0.298 | 0 (0.0) | - | - |
| | Allele | A | 239 (73.8) | 437 (77.5) | Ref | | 162 (71.7) | Ref | | 19 (79.2) | Ref | |
| | | C | 85 (26.2) | 127 (22.5) | 1.31 (0.93-1.85) | 0.214 | 64 (28.3) | 1.26 (0.84-1.9) | 0.430 | 5 (20.8) | 0.67 (0.16-2.11) | 0.941 |
| | Dominant | AA | 91 (56.2) | 170 (60.3) | Ref | | 61 (54.0) | Ref | | 7 (58.3) | Ref | |
| | | CC + CA | 71 (43.8) | 112 (39.7) | 1.75 (0.76-3.97) | 0.270 | 52 (46.0) | 1.99 (0.76-4.99) | 0.298 | 5 (41.7) | 0.7 (0.14-2.81) | 0.941 |
| | Recessive | CA + AA | 148 (91.4) | 267 (94.7) | Ref | | 101 (89.4) | Ref | | 12 (100.0) | - | |
| | | CC | 14 (8.6) | 15 (5.3) | 1.3 (0.85-2) | 0.307 | 12 (10.6) | 1.18 (0.71-1.96) | 0.679 | 0 (0.0) | - | - |
| | Over-dominant | CC + AA | 105 (64.8) | 185 (65.6) | Ref | | 73 (64.6) | Ref | | 7 (58.3) | Ref | |
| | | CA | 57 (35.2) | 97 (34.4) | 0.88 (0.57-1.37) | 0.577 | 40 (35.4) | 1.03 (0.61-1.76) | 0.908 | 5 (41.7) | 1.18 (0.29-5.92) | 0.941 |
| **rs1800871 -819C/T** | Genotypes | TT | 92 (56.8) | 178 (63.1) | Ref | | 60 (53.1) | Ref | | 8 (66.7) | Ref | |
| | | CT | 56 (34.6) | 91 (32.3) | 1.25 (0.8-1.97) | 0.392 | 41 (36.3) | 1.23 (0.72-2.1) | 0.622 | 4 (33.3) | 0.89 (0.18-3.62) | 0.941 |
| | | CC | 14 (8.6) | 13 (4.6) | 2.17 (0.92-5.15) | 0.193 | 12 (10.6) | 2.47 (0.9-6.61) | 0.188 | 0 (0.0) | - | - |
| | Allele | T | 240 (74.1) | 447 (79.3) | Ref | | 161 (71.2) | Ref | | 20 (83.3) | Ref | |
| | | C | 84 (25.9) | 117 (20.7) | 1.45 (1.02-2.06) | 0.193 | 65 (28.8) | 1.49 (1.08-2.25) | 0.173 | 4 (16.7) | 0.77 (0.18-2.5) | 0.941 |
| | Dominant | TT | 92 (56.8) | 178 (63.1) | Ref | | 60 (53.1) | Ref | | 8 (66.7) | Ref | |
| | | CC + CT | 70 (43.2) | 104 (36.9) | 2.16 (0.92-5.09) | 0.193 | 53 (46.9) | 2.58 (0.94-6.89) | 0.177 | 4 (33.3) | 0.82 (0.16-3.35) | 0.941 |
| | Recessive | CT + TT | 148 (91.4) | 269 (95.4) | Ref | | 101 (89.4) | Ref | | 12 (100.0) | - | |
| | | CC | 14 (8.6) | 13 (4.6) | 1.43 (0.93-2.2) | 0.202 | 12 (10.6) | 1.43 (0.86-2.39) | 0.299 | 0 (0.0) | - | - |
| | Over-dominant | CC + TT | 106 (65.4) | 191 (67.7) | Ref | | 72 (63.7) | Ref | | 8 (66.7) | Ref | |
| | | CT | 56 (34.6) | 91 (32.3) | 0.84 (0.54-1.31) | 0.464 | 41 (36.3) | 0.88 (0.52-1.5) | 0.702 | 4 (33.3) | 1.06 (0.26-5.32) | 0.941 |
| | | | **DF n = 158** | **Control n = 234** | | | **DWS n = 112** | | | **SD n = 12** | | |
| **rs1800896 -1082G/A** | Genotypes | AA | 131 (82.9) | 64 (27.4) | | | 95 (84.8) | | | 12 (100.0) | | |
| | | GA | 23 (14.6) | 1 (0.4) | 0.53 (0.29-0.91) | 0.193 | 16 (14.3) | 0.39 (0.19-0.76) | **0.049** | 0 (0.0) | - | - |
| | | GG | 4 (2.5) | 169 (72.2) | 8.24 (1.07-171.34) | 0.193 | 1 (0.9) | 0.18 (0-9.2) | 0.606 | 0 (0.0) | - | - |
| | Allele | A | 285 (90.2) | 129 (27.6) | | | 206 (92.0) | | | 24 (100.0) | | |
| | | G | 31 (9.8) | 339 (72.4) | 0.78 (0.48-1.26) | 0.392 | 18 (8.0) | 0.44 (0.22-0.81) | **0.049** | 0 (0.0) | - | - |
| | Dominant | AA | 131 (82.9) | 64 (27.4) | | | 95 (84.8) | | | 12 (100.0) | | |
| | | GG + GA | 27 (17.1) | 170 (72.6) | 10.58 (1.39-218.45) | 0.193 | 17 (15.2) | 0.31 (0-19.03) | 0.702 | 0 (0.0) | - | - |
| | Recessive | GA + AA | 154 (97.5) | 65 (27.8) | | | 111 (99.1) | | | 12 (100.0) | | |
| | | GG | 4 (2.5) | 169 (72.2) | 0.63 (0.36-1.06) | 0.196 | 1 (0.9) | 0.4 (0.19-0.77) | **0.049** | 0 (0.0) | - | - |
| | Over-dominant | GG + AA | 135 (85.4) | 233 (99.6) | | | 96 (85.7) | | | 12 (100.0) | | |
| | | GA | 23 (14.6) | 1 (0.4) | 1.96 (1.13-3.49) | 0.193 | 16 (14.3) | 2.48 (1.28-5.09) | **0.049** | 0 (0.0) | - | - |

Genotype and allele distribution of *IL-10* polymorphisms in dengue infected patients with different DENV serotypes were investigated (Table 4). No significant difference was found between compared groups.

**Table 4. Genotype and allele distribution of *IL-10* polymorphisms in dengue patients with different DENV serotypes and healthy controls. Data are shown in number and percentage. Bold values indicate statistical significance. Odds ratio and p-value were calculated by using binary logistic regression model and adjusted for age and sex. P-values were adjusted by the Benjamini-Hochberg procedure.**

| SNP | Variable | Genotypes | DENV-1 | Control | OR (95% CI) | p-value | DENV-2 | OR 2 (95% CI) | p-value | DENV-4 | OR 3 (95% CI) | p-value |
|---|---|---|---|---|---|---|---|---|---|---|---|---|
| **rs1800872 -592C/A** | | | n = 105 | n = 282 | | | n = 199 | | | n = 19 | | |
| | Genotypes | AA | 61 (58.1) | 170 (60.3) | Ref | | 111 (55.8) | Ref | | 9 (47.4) | Ref | |
| | | CA | 33 (31.4) | 97 (34.4) | 1.04 (0.61-1.75) | 0.967 | 70 (35.2) | 1.18 (0.76-1.81) | 0.458 | 9 (47.4) | 1.31 (0.44-3.74) | 0.781 |
| | | CC | 11 (10.5) | 15 (5.3) | 1.78 (0.7-4.37) | 0.360 | 18 (9.0) | 2.05 (0.92-4.62) | 0.160 | 1 (5.2) | 1.54 (0.08-9.64) | 0.781 |
| | Allele | A | 155 (73.8) | 437 (77.5) | Ref | | 292 (73.4) | Ref | | 27 (71.1) | Ref | |
| | | C | 55 (26.2) | 127 (22.5) | 1.22 (0.82-1.8) | 0.456 | 106 (26.6) | 1.34 (0.96-1.87) | 0.160 | 11 (28.9) | 1.2 (0.51-2.62) | 0.781 |
| | Dominant | AA | 61 (58.1) | 170 (60.3) | Ref | | 111 (55.8) | Ref | | 9 (47.4) | Ref | |
| | | CC + CA | 44 (41.9) | 112 (39.7) | 1.15 (0.7-1.87) | 0.753 | 88 (44.2) | 1.31 (0.87-1.98) | 0.247 | 10 (52.6) | 1.25 (0.44-3.5) | 0.781 |
| | Recessive | CA + AA | 94 (89.5) | 267 (94.7) | Ref | | 181 (91.0) | Ref | | 18 (94.7) | Ref | |
| | | CC | 11 (10.5) | 15 (5.3) | 1.74 (0.7-4.19) | 0.360 | 18 (9.0) | 1.92 (0.88-4.24) | 0.177 | 1 (5.3) | 1.35 (0.07-7.93) | 0.781 |
| | Over-dominant | CC + AA | 72 (68.6) | 185 (65.6) | Ref | | 129 (64.8) | Ref | | 10 (52.6) | Ref | |
| | | CA | 33 (31.4) | 97 (34.4) | 1.03 (0.62-1.74) | 0.967 | 70 (35.2) | 0.9 (0.59-1.38) | 0.636 | 9 (47.4) | 0.84 (0.3-2.52) | 0.781 |
| **rs1800871 -819C/T** | | | n = 105 | n = 282 | | | n = 199 | | | n = 19 | | |
| | Genotypes | TT | 62 (59.0) | 178 (63.1) | Ref | | 111 (55.8) | Ref | | 9 (47.4) | Ref | |
| | | CT | 31 (29.5) | 91 (32.3) | 1.08 (0.63-1.83) | 0.931 | 69 (34.7) | 1.29 (0.83-1.99) | 0.308 | 9 (47.4) | 1.35 (0.45-3.88) | 0.781 |
| | | CC | 12 (11.4) | 13 (4.6) | 2.41 (0.95-6.04) | 0.201 | 19 (9.5) | 2.42 (1.08-5.57) | 0.119 | 1 (5.2) | 1.83 (0.09-11.72) | 0.781 |
| | Allele | T | 155 (73.8) | 447 (79.3) | Ref | | 291 (73.1) | Ref | | 27 (71.1) | Ref | |
| | | C | 55 (26.2) | 117 (20.7) | 1.43 (0.95-2.12) | 0.211 | 107 (26.9) | 1.53 (1.09-2.14) | 0.090 | 11 (28.9) | 1.38 (0.58-3.03) | 0.781 |
| | Dominant | TT | 62 (59.0) | 178 (63.1) | Ref | | 111 (55.8) | Ref | | 9 (47.4) | Ref | |
| | | CC + CT | 43 (41.0) | 104 (36.9) | 1.31 (0.79-2.14) | 0.434 | 88 (44.2) | 1.5 (0.99-2.27) | 0.139 | 10 (52.6) | 1.45 (0.51-4.07) | 0.781 |
| | Recessive | CT + TT | 93 (88.6) | 269 (95.4) | Ref | | 180 (90.5) | Ref | | 18 (94.7) | Ref | |
| | | CC | 12 (11.4) | 13 (4.6) | 2.54 (1.01-6.3) | 0.201 | 19 (9.5) | 2.41 (1.08-5.5) | 0.119 | 1 (5.3) | 1.93 (0.1-11.96) | 0.781 |
| | Over-dominant | CC + TT | 74 (70.5) | 191 (67.7) | Ref | | 130 (65.3) | Ref | | 10 (52.6) | Ref | |
| | | CT | 31 (29.5) | 91 (32.3) | 1 (0.59-1.69) | 0.989 | 69 (34.7) | 0.83 (0.54-1.27) | 0.440 | 9 (47.4) | 0.77 (0.27-2.29) | 0.781 |
| **rs1800896 -1082G/A** | | | n = 101 | n = 234 | | | n = 195 | | | n = 19 | | |
| | Genotypes | AA | 86 (85.1) | 169 (72.2) | Ref | | 162 (83.1) | Ref | | 16 (84.2) | Ref | |
| | | GA | 13 (12.9) | 64 (27.4) | 0.46 (0.22-0.89) | 0.201 | 29 (14.9) | 0.52 (0.3-0.87) | 0.090 | 3 (15.8) | 0.59 (0.13-1.97) | 0.781 |
| | | GG | 2 (2.0) | 1 (0.4) | 7.88 (0.65-186.15) | 0.254 | 4 (2.1) | 5.41 (0.57-122.13) | 0.246 | 0 (0.0) | – | – |
| | Allele | A | 185 (91.6) | 402 (85.9) | Ref | | 353 (90.5) | Ref | | 35 (92.1) | Ref | |
| | | G | 17 (8.4) | 66 (14.1) | 0.68 (0.37-1.2) | 0.360 | 37 (9.5) | 0.7 (0.43-1.11) | 0.195 | 3 (7.9) | 0.63 (0.15-1.91) | 0.781 |
| | Dominant | AA | 86 (85.1) | 169 (72.2) | Ref | | 162 (83.1) | Ref | | 16 (84.2) | Ref | |
| | | GG + GA | 15 (14.9) | 65 (27.8) | 0.54 (0.27-1.02) | 0.201 | 33 (16.9) | 0.58 (0.35-0.97) | 0.177 | 3 (15.8) | 0.61 (0.13-2.02) | 0.781 |
| | Recessive | GA + AA | 99 (98.0) | 233 (99.6) | Ref | | 191 (97.9) | Ref | | 19 (100.0) | Ref | |
| | | GG | 2 (2.0) | 1 (0.4) | 10.59 (0.9-247.04) | 0.201 | 4 (2.1) | 7.27 (0.79-161.08) | 0.120 | 0 (0.0) | – | – |
| | Over-dominant | GG + AA | 88 (87.1) | 170 (72.6) | Ref | | 166 (85.1) | Ref | | 16 (84.2) | Ref | |
| | | GA | 13 (12.9) | 64 (27.4) | 2.21 (1.14-4.54) | 0.201 | 29 (14.9) | 1.96 (1.16-3.37) | 0.090 | 3 (15.8) | 1.61 (0.48-7.32) | 0.781 |

**Table 5. Association of *IL-10* haplotypes and dengue. The alleles in the haplotype were ranked in the SNP order of rs1800896, rs1800871 and rs1800872. Associations were tested using logistic regression adjusted for age and sex.**

| Haplotypes | Number of observation (%) | | OR (95% CI) | p-value |
|---|---|---|---|---|
| | DENV (n=574) | Control (n=564) | | |
| ATA | 402 (70.0%) | 394 (69.8%) | Ref | |
| GTA | 12 (2.0%) | 38 (6.7%) | 0.31 (0.14 - 0.67) | 0.003 |
| ACC | 108 (19.0%) | 78 (14.0%) | 1.36 (0.99 - 1.89) | 0.061 |
| GCC | 34 (6.0%) | 34 (6.4%) | 1.08 (0.64 - 1.82) | 0.783 |
| Minor | 18 (3.0%) | 20 (3.1%) | 0.66 (0.35 - 1.22) | 0.182 |

= 0.061) No significant difference in IL-10 serum levels between varying haplotypes was found (Fig 3).

## Discussion

The study investigated host and viral genetic factors for dengue susceptibility and severity. We report the incidence of dengue in two consecutive outbreaks in 2021-2022 and showed that DENV-1 and DENV-2 were the predominant serotypes in circulation and distributed among severe cases. Plasma levels of the host cytokines IL-10 were significantly elevated in dengue patients of varying severity. The promoter variants of *IL-10*, which have been described to regulate IL-10 levels [13], were also differentially distributed in dengue patients and in healthy controls.

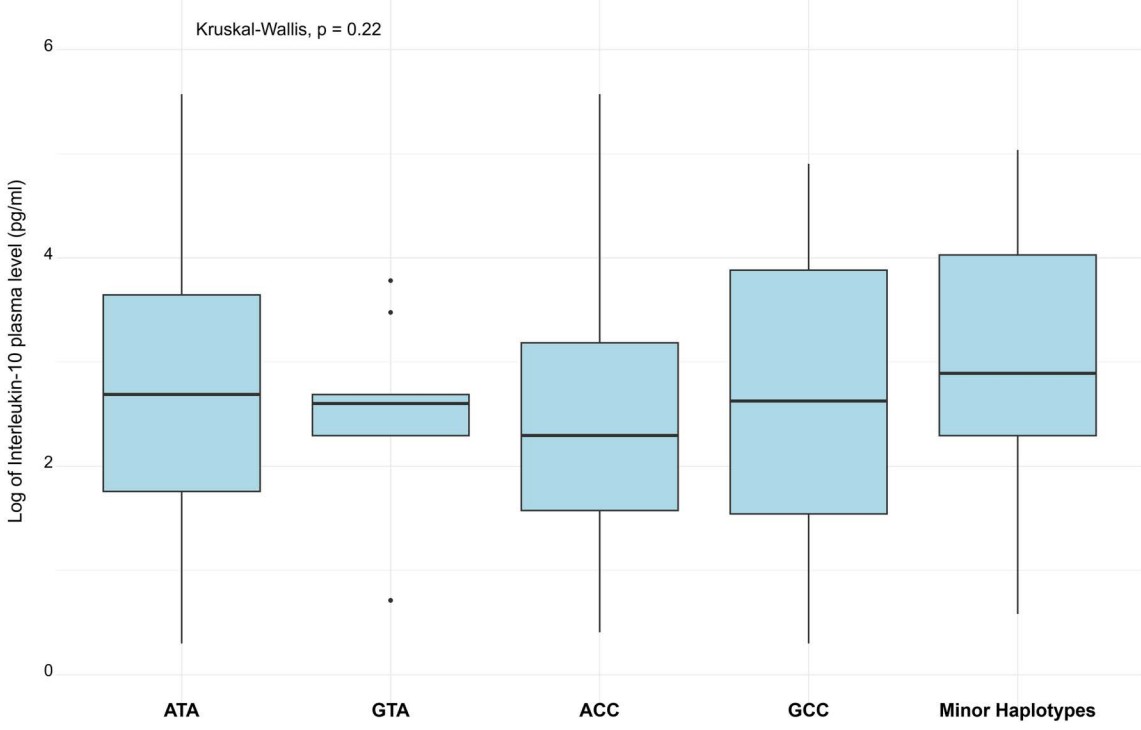

**Fig 3. IL-10 plasma levels in different IL-10 haplotypes.**

The main DENV serotypes circulating during the 2021-2022 outbreak were DENV-2 and DENV-1, followed by DENV-4. In northern Vietnam, the distribution of DENV-1,-2 serotypes have undergone changes in recent years. DENV-1 was previously observed as the predominant serotype during the years 2017-2019 [18]. However, during the outbreaks in 2019-2020, there was a shift to DENV-2 dominance [19], a trend that persisted throughout the 2021-2022 period, as observed in our study. The DENV-1,-2,-4 serotypes were persisting in the northern population compared to other regions, where all four serotypes co-existed [20]. It is well described that DENV-2 is associated with severe disease progression [2], and more than 61% of DENV-2 infections were observed in patients with DWS and severe dengue in our study. This finding is consistent with data from previous outbreaks that posed an increased burden on public health [21]. While DENV-3 serotype were not observed in our study, this appears to be more regional in Vietnam, as reported in previous outbreaks [20].

The host cytokine plasma IL-10, which is well described as an anti-inflammatory marker in dengue [22], was significantly associated to dengue severity. While patients without warning signs had lower IL-10 plasma levels, higher levels were indicative of disease deterioration, as observed in severe cases. Elevated IL-10 levels in dengue pathogenesis have been shown to reflect a reduced type I interferon response and thus a delayed viral clearance during acute episodes [22]. During DENV infection, there is a reduction in IFN-γ production by T cells, and an early increase in IL-10 production, leading to less IFN- γ and more viral persistence, as shown from other viral infections [9,23].

Exacerbated inflammatory response and altered vascular function are hallmarks of dengue [24]. IL-10 acts as an immunoregulatory cytokine in both the Th1 and Th2 responses to maintain immune homeostasis and to prevent excessive inflammation and tissue damage. While an optimal IL-10 response might mitigate immune-mediated pathology, an excessive IL-10 milieu could hinder effective viral clearance and potentially aggravate the clinical course of the disease.

Upon monocyte infection with DENV, the expression of *IL-10* gradually rises over time and more in the case of ADE [22]. Secondary infections trigger an earlier and higher release of IL-6 and IL-10 compared to primary infections, indicating that pre-existing immune memory accelerates cytokine production in subsequent infections [25]. Several transcription factors are involved in the modulation of *IL-10* transcription, and approximately 75% of the variation in IL-10 secretion capacity derives from genetic factors [26,27]. Further, a study has demonstrated a strong genetic influence on cytokine production, including IL-10, upon ex vivo stimulation [28]. This suggests that genetic factors may influence IL-10 levels, thereby affecting the severity and susceptibility to dengue. However, no significant difference in IL-10 plasma levels was observed between the genetic variants in our study. It is also important to note that dengue pathogenesis is complex and involves multiple factors, ranging from the infected viral serotype, sequence of infections (primary/secondary) to the dynamics of other immune modulators upon infection. Therefore, anticipating the level of IL-10 in dengue patients is complex and dependent on multiple factors.

Nonetheless, an earlier study found that ACC haplotype (-1082A/-819C/-592C) is associated with downregulated IL-10 level in dengue patients [29], which consistent with our preliminary observation, but the results from our study did not reach the statistical significance level (Fig 3). The study findings show a marked association between *IL-10* promoter rs1800896 (-1082G/A) and rs1800871 (-819C/T) with DENV infection. Allelic frequencies observed in our study align with data published on the Ensembl database (https://www.ensembl.org/index.html) for the Kinh ethnicity (KHV): rs1800896 (A>G), rs1800871 (T>C), and rs1800872 (A>C).

Our observations suggest that the allele -819C correlates with a significantly increased risk of DENV infection compared to -819T. On the other hand, the genotype -1082GA and allele -1082G were found to be protective factors against developing dengue and DWS. Haplotype analysis similarly revealed that the GTA haplotype (-1082G/-819T/-592A) conferred protection from dengue in the Vietnamese population. Aligned with our findings, the GTA haplotype was reported to be associated with protection from other infectious diseases such as schistosomiasis and malaria in the Nigerian population [15] and in the Brazilian population [30]. Considering that the predominant IL-10 genetic variant in the Vietnamese population is ATA (-1082A/-819T/-592A), with the GTA haplotype being less frequent, the high susceptibility and burden of dengue observed in the Vietnamese population may be partially due to genetic characteristics of the Vietnamese population.

In comparison to a recent study from Mexico, where the allelic distribution is different from Vietnam (Mexico: rs1800896 (A<G), rs1800871 (T<C), and rs1800872 (A<C); https://www.ensembl.org/index.html), we found that the distribution of the SNPs rs1800872 (-592C/A) and rs1800871 (-819C/T) among DENV infected patients are different. A study found that -819T and -592A increase the risk of DENV infection in the Mexican population [27], while our study revealed that these two alleles appear to have a protective role against DENV infection in the Vietnamese population. However, given that the allelic distributions of the three studied SNPs are opposite in these two populations, the association of *IL-10* genetic variants with dengue might be influenced, thus affecting the generalization of the study findings.

Although our study is the first to investigate the association between *IL-10* SNPs and DENV infection in the Vietnamese population, a relatively large sample size would be helpful for further studies to justify our findings. While a significant association between IL-10 levels of varying severity and the association of promoter *IL-10* variants was observed in our study, other cytokines and chemokines may also regulate the course of infection. Further studies are warranted, such as the explicit investigation of a large panel of cytokines and chemokines to understand how they are co-regulated during dengue infection, and if so, how they are modulated during the clinical course of DENV infection.

In summary, this study underscores the importance of understanding genetic predispositions and immune response mechanisms in dengue severity, particularly in endemic regions like Vietnam. The identification of IL-10 plasma levels and genetic variants, specifically the protective role of the -1082G allele, provides valuable insights into host-pathogen interactions and susceptibility to severe dengue. Such findings can guide the development of targeted public health strategies, including early risk stratification, personalized treatment approaches, and the design of immunogenetic-based interventions. Furthermore, the consistent dominance of DENV-2 and the associated higher severity highlights the need for serotype-specific monitoring and vaccination campaigns to mitigate the growing burden of dengue in Southeast Asia. This integrative approach, combining clinical, genetic, and serotype data, is pivotal in enhancing preparedness and response efforts in dengue-endemic regions.

## Acknowledgements

We thank all study subjects for their participation. We acknowledge the support from the Open Access Publication Fund of the University of Tübingen.

## Author contributions

**Conceptualization:** Jonas Schmidt-Chanasit, Peter G Kremsner, Le Huu Song, Thirumalaisamy P. Velavan.

**Data curation:** Nguyen Trong The, Truong Nhat My, Nguyen Linh Toan, Le Huu Song.

**Formal analysis:** Do Duc Anh, Lara Vugrek.

**Funding acquisition:** Thirumalaisamy P. Velavan.

**Investigation:** Do Duc Anh, Lara Vugrek, Nourhane Hafza, Le Thi Kieu Linh.

**Methodology:** Do Duc Anh, Lara Vugrek, Nourhane Hafza, Le Thi Kieu Linh.

**Project administration:** Peter G Kremsner, Thirumalaisamy P. Velavan.

**Resources:** Peter G Kremsner, Le Huu Song, Thirumalaisamy P. Velavan.

**Software:** Do Duc Anh.

**Supervision:** Thirumalaisamy P. Velavan.

**Validation:** Do Duc Anh, Le Thi Kieu Linh, Do Huy Loc, Jonas Schmidt-Chanasit, Thirumalaisamy P. Velavan.

**Visualization:** Do Duc Anh, Do Huy Loc.

**Writing – original draft:** Do Duc Anh, Thirumalaisamy P. Velavan.

**Writing – review & editing:** Do Duc Anh, Truong Nhat My, Jonas Schmidt-Chanasit, Peter G Kremsner, Le Huu Song, Thirumalaisamy P. Velavan.

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
