## [Decision Letter · Decision Letter 0]

17 Jan 2025

PNTD-D-24-00849Characterization of Dengue Patients in Vietnam: Clinical, Virological, and IL-10 Profiles during 2021- 2022 outbreaksPLOS Neglected Tropical DiseasesDear Dr. Velavan, Thank you for submitting your manuscript to PLOS Neglected Tropical Diseases. After careful consideration, we feel that it has merit but does not fully meet PLOS Neglected Tropical Diseases's publication criteria as it currently stands. Therefore, we invite you to submit a revised version of the manuscript that addresses the points raised during the review process. Please submit your revised manuscript within 30 days Mar 18 2025 11:59PM. If you will need more time than this to complete your revisions, please reply to this message or contact the journal office at plosntds@plos.org. Please include the following items when submitting your revised manuscript:

* A rebuttal letter that responds to each point raised by the editor and reviewer(s). You should upload this letter as a separate file labeled 'Response to Reviewers '. This file does not need to include responses to any formatting updates and technical items listed in the 'Journal Requirements' section below. * A marked-up copy of your manuscript that highlights changes made to the original version. You should upload this as a separate file labeled 'Revised Manuscript with Track Changes '.

* An unmarked version of your revised paper without tracked changes. You should upload this as a separate file labeled 'Manuscript '.

We look forward to receiving your revised manuscript.

Kind regards,

Adly M.M. Abd-Alla, Prof asso.

Academic Editor

Abdallah Samy

Section Editor

Shaden Kamhawi

co-Editor-in-Chief

Paul Brindley

co-Editor-in-Chief

**Additional Editor Comments:** Thanks very much for considering PLOS Neglected Tropical Diseases for your submission. Your manuscript received three reviews; all reviews suggested some comments that should be addressed before considering a revised version of your manuscript. Please review carefully all the comments and suggestions below before submitting a revised version of your manuscript. **Journal Requirements:**

At this stage, the following Authors/Authors require contributions: Do Duc Anh, Lara Vugrek, Nguyen Trong The, Nourhane Hafza, Truong Nhat My, Le Thi Kieu Linh, Do Huy Loc, Jonas Schmidt-Chanasit, Nguyen Linh Toan, Peter G Kremsner, Le Huu Song, and Thirumalaisamy P. Velavan. Please ensure that the full contributions of each author are acknowledged in the "Add/Edit/Remove Authors" section of our submission form.

- ® on page: 7.

4) We note that your Data Availability Statement is currently as follows: "All data is available within the manuscript.". Please confirm at this time whether or not your submission contains all raw data required to replicate the results of your study. Authors must share the “minimal data set” for their submission. PLOS defines the minimal data set to consist of the data required to replicate all study findings reported in the article, as well as related metadata and methods (https://journals.plos.org/plosone/s/data-availability#loc-minimal-data-set-definition).

- The points extracted from images for analysis..

5) Please ensure that the funders and grant numbers match between the Financial Disclosure field and the Funding Information tab in your submission form. Note that the funders must be provided in the same order in both places as well.

**Reviewers' comments:**

Reviewer's Responses to Questions

**Key Review Criteria Required for Acceptance?**

**Methods**

-Are the objectives of the study clearly articulated with a clear testable hypothesis stated?

-Is the study design appropriate to address the stated objectives?

-Is the population clearly described and appropriate for the hypothesis being tested?

-Is the sample size sufficient to ensure adequate power to address the hypothesis being tested?

-Were correct statistical analysis used to support conclusions?

-Are there concerns about ethical or regulatory requirements being met?

Reviewer #1: The objective of the study is clearly stated, and the population well described. There is no ethical concern and the statical analysis was well presented. The authors did not show how they arrived at the sample size. I assumed convenient sampling method was employed. The author should state clearly the sampling technique that was used.

Reviewer #2: This paper describes an important topic that is growing in impact as numbers of dengue cases are rising in many disparate geographic areas of the globe. Its strengths include the numbers of cases studied, the inclusion of clinically severe cases and the detailed analysis of Il-10 promotor polymorphism associations with disease severity allowing for dengue serotypes. An additional strength is the exclusion of infection by closely related flaviviruses.

**Results**

-Does the analysis presented match the analysis plan?

-Are the results clearly and completely presented?

-Are the figures (Tables, Images) of sufficient quality for clarity?

Reviewer #1: The analysis plan is good and well presented. The figures and tables are sufficient. The author might consider removing figure 1b because same data is well presented in Table 1.

Reviewer #2: The analysis of associations for polymorphisms using logistic regression models and different assignments of gene transmission (*dominant, recessive and so on) is appropriate, and the findings are of interest.

The results are clearly presented.

**Conclusions**

-Are the conclusions supported by the data presented?

-Are the limitations of analysis clearly described?

-Do the authors discuss how these data can be helpful to advance our understanding of the topic under study?

-Is public health relevance addressed?

Reviewer #1: The conclusion was clear, and the study was well discussed but the public health implication is not clearly stated

Reviewer #2: As mentioned there may be other chemo-cytokines that are relevant to this topic and further comments on this topic may be helpful for context, including in the discussion. Are there any in vitro data looking at Il10 promotor polymorphisms in ex vivo experiments that might shed light on the functional relevance of these polymorphisms? For example, on the timing of release of Il-10 (PMCID: PMC11508614 DOI: 10.3390/ijms252011238) and with relevance to another earlier model of infection, which the authors might consider worth mentioning in discussion (reference abstract appended).

Lancet

. 1997 Jan 18;349(9046):170-3. doi: 10.1016/s0140-6736(96)06413-6.

Genetic influence on cytokine production and fatal meningococcal disease

R G Westendorp 1 , J A Langermans, T W Huizinga, A H Elouali, C L Verweij, D I Boomsma, J P Vandenbroucke

Affiliations expand

PMID: 9111542 DOI: 10.1016/s0140-6736(96)06413-6

Erratum in

Lancet 1997 Mar 1;349(9052):656. Vandenbrouke JP [corrected to Vandenbroucke JP]

Abstract

Background: To assess the genetic influence on cytokine production and its contribution to fatal outcome, we determined the capacity to produce tumour necrosis factor-alpha (TNF alpha) and interleukin-10 (IL-10) in families of patients who had had meningococcal disease.

Methods: We studied 190 first-degree relatives of 61 patients with meningococcal disease; we also studied 26 monozygotic twins. Production of cytokines was determined during endotoxin stimulation of whole-blood samples ex-vivo. Heritability was estimated in a pedigree-based maximum-likelihood model. DNA was typed for the G to A transition polymorphisms at position -308 and -238 in the TNF gene promoter.

Findings: Heritability in monozygotic twins was 0.60 for the production of TNF and 0.75 for the production of IL-10. Families with low TNF production had a tenfold increased risk for fatal outcome (OR 8.9, 95% CI 1.8-45), whereas high IL-10 production increased the risk 20-fold (19.5, 2.3-165). Families with both characteristics had the greatest risk. The transition polymorphisms in the TNF gene promoter were not associated with outcome.

Interpretation: Genetic factors substantially influence production of cytokines. An innate anti-inflammatory cytokine profile may contribute to fatal meningococcal disease.

**Editorial and Data Presentation Modifications?**

Reviewer #1: (No Response)

Reviewer #2: Minor Revision

**Summary and General Comments**

Reviewer #1: Overall the study is well executed and the paper well written

The inheritance model analysis section should be part of the statistical analysis.

Reviewer #2: The paper is well written.

PLOS authors have the option to publish the peer review history of their article (what does this mean? ). If published, this will include your full peer review and any attached files.

**Do you want your identity to be public for this peer review?** For information about this choice, including consent withdrawal, please see our Privacy Policy .

Reviewer #1: No

Reviewer #2: No

**Figure resubmission:** While revising your submission, please upload your figure files to the Preflight Analysis and Conversion Engine (PACE) digital diagnostic tool, https://pacev2.apexcovantage.com/. PACE helps ensure that figures meet PLOS requirements. To use PACE, you must first register as a user. Registration is free. Then, login and navigate to the UPLOAD tab, where you will find detailed instructions on how to use the tool. If you encounter any issues or have any questions when using PACE, please email PLOS at figures@plos.org. Please note that Supporting Information files do not need this step. If there are other versions of figure files still present in your submission file inventory at resubmission, please replace them with the PACE-processed versions.**Reproducibility:** To enhance the reproducibility of your results, we recommend that authors of applicable studies deposit laboratory protocols in protocols.io, where a protocol can be assigned its own identifier (DOI) such that it can be cited independently in the future. Additionally, PLOS ONE offers an option to publish peer-reviewed clinical study protocols. Read more information on sharing protocols at https://plos.org/protocols?utm_medium=editorial-email&utm_source=authorletters&utm_campaign=protocols

---

## [Decision Letter · Decision Letter 1]

28 Feb 2025

Dear Prof. Dr Velavan,

We are pleased to inform you that your manuscript 'Characterization of Dengue Patients in Vietnam: Clinical, Virological, and IL-10 Profiles during 2021- 2022 outbreaks' has been provisionally accepted for publication in PLOS Neglected Tropical Diseases.

Best regards,

Adly M.M. Abd-Alla, Prof asso.

Academic Editor

Abdallah Samy

Section Editor

Shaden Kamhawi

co-Editor-in-Chief

Paul Brindley

co-Editor-in-Chief

---

## [Editor Report · Acceptance letter]

Dear Prof. Dr Velavan,

We are delighted to inform you that your manuscript, "Characterization of Dengue Patients in Vietnam: Clinical, Virological, and IL-10 Profiles during 2021- 2022 outbreaks," has been formally accepted for publication in PLOS Neglected Tropical Diseases.

Best regards,

Shaden Kamhawi

co-Editor-in-Chief

Paul Brindley

co-Editor-in-Chief
